# Vitamin D Ameliorates the Hepatic Oxidative Damage and Fibrotic Effect Caused by Thioacetamide in Rats

**DOI:** 10.3390/biomedicines11020424

**Published:** 2023-02-01

**Authors:** Aya Megahed, Hossam Gadalla, Fatma M. Abdelhamid, Samah J. Almehmadi, Anmar A. Khan, Talat A. Albukhari, Engy F. Risha

**Affiliations:** 1Department of Clinical Pathology, Faculty of Veterinary Medicine, Mansoura University, Mansour 35516, Egypt; 2Laboratory Medicine Department, Faculty of Applied Medical Sciences, Umm Al-Qura University, Al Abdeyah, Makkah P.O. Box 7607, Saudi Arabia; 3Department of Immunology and Hematology, Faculty of Medicine, Umm Al-Qura University, Makkah P.O. Box 7607, Saudi Arabia

**Keywords:** vitamin D_3_, thioacetamide, hepatic oxidative stress, fibrosis scoring, fibrotic markers

## Abstract

Vitamin D_3_ (VD_3_) is a sunshine hormone that regulates cellular proliferation, differentiation, apoptosis, and angiogenesis related to liver parenchyma. We used a thioacetamide (TAA)-induced hepatic fibrosis rat model in our study to investigate the beneficial roles of VD_3_ to overcome extensive liver fibrosis. Randomly, four equal groups (eight rats per group) underwent therapy for eight successive weeks: a control group, a group treated with TAA 100 mg/kg BW IP every other day, a group treated with VD_3_ 1000 IU/kg BW IM every day, and a TAA+VD group treated with both therapies. Treatment with VD_3_ after TAA-induced hepatic fibrosis was found to alleviate elevated liver function measures by decreasing ALT, AST, and ALP activity; decreasing total bilirubin, direct bilirubin, cholesterol, and triglyceride levels; and increasing glucose and 25[OH]D_3_. Rats treated with VD_3_ showed marked decreases in MDA and increased SOD, CAT, and GSH levels. In addition, CD34 and FGF23 gene expressions were reduced after dual therapy. Liver sections from the TAA+VD group showed markedly decreased hepatic lesions, and Masson’s trichrome stain showed a marked decrease in dense bluish-stained fibrous tissue. The immunohistochemical expression of TGF-β and α-SMA showed markedly decreased positive brown cytoplasmic expression in a few hepatocytes, clarifying the antifibrotic effect of VD_3_ in hepatic fibrosis. In conclusion, VD_3_ alleviates hepatotoxicity and fibrosis caused by TAA.

## 1. Introduction

The regulation of various physiological processes, including the biochemical metabolisms of protein, carbohydrate, and fat for growth, nutrient supply, energy provision, detoxification, bile secretion, vitamin storage, fighting against various diseases, and reproductive functions, are modulated within hepatocytes [1]. All liver cell types make their own contribution to the development of liver fibrosis, with cross-talk between cells of different types in the process of fibrosis through the release of specific mediators such as interleukins, growth factors, and reactive oxygen species (ROS) [2,3].

During the course of liver injury, some protective molecular pathways are repressed. One of these important signaling pathways is the cyclic adenosine monophosphate (cAMP) pathway. cAMP is a second messenger that plays a central role in cellular responses to neurotransmitters and hormones and regulates various cellular functions, including inflammation and cell differentiation, by affecting gene/protein expression and function [4,5]. Increased cAMP levels have been found to inhibit the conversion of resting fibroblasts/fibroblast-like cells, such as hepatic stellate cells (HSC), to profibrogenic myofibroblasts after cell injury, decrease their proliferation, stimulate their death, and inhibit extracellular matrix (ECM) protein synthesis, which causes the destruction of the natural liver architecture and the emergence of liver fibrosis [4,6,7].

Thioacetamide (C_2_H_5_NS) is an organosulfur compound, to which occupational exposure usually occurs through inhalation and dermal contact at the workplace [8]. TAA is extensively used in the development of suitable animal models for acute and chronic liver injury, with various doses, times, and routes of administration employed to resemble human liver fibrosis and cirrhosis [9]. Hepatotoxicity with TAA progresses when it is converted by the cytochrome P450 enzyme (CYP450) into TAA–sulfur dioxide (TAA-S-dioxide), an unstable reactive metabolite that initiates necrosis and ROS generation by binding covalently to liver macromolecules and liver parenchyma and invasion by inflammatory cells, causing centrilobular necrosis and activating HSCs [10,11].

The active form of vitamin D is 1, 25-dihydroxy cholecalciferol (1, 25[OH]_2_D_3_); it not only supports human bone health, but also directly or indirectly manages many genes controlling cellular division, differentiation, apoptosis, and angiogenesis [12,13]. Many conflicting recent studies are showing an association between vitamin D (VD) deficiency and cancer, cardiovascular disease, diabetes, autoimmune diseases, and depression [14,15]. It has also been shown to control free radical generation in hepatocytes of diabetic mice through binding to the VDR in the cell nucleus [16], reducing the risk or severity of chronic liver disorders [17]. Recently, VD_3_ has drawn more attention as a non-enzymatic antioxidant, with physiological concentrations having a higher antioxidant effect than vitamin E, melatonin, and β-estradiol [18]. VD_3_ exerts its action through a mechanism involved in reducing the level of lipid peroxidation and increasing the total antioxidant capacity in induced muscle proteolysis [19] and streptozotocin-induced diabetes [20].

According to the WHO, identifying natural products as non-drug remedies for numerous diseases is of major interest due to the lack of effective medications for the treatment of liver diseases [21]. Regrettably, there are no studies on the hepatoprotective effect of VD_3_ against TAA-induced hepatotoxicity and fibrosis. Therefore, the present study was designed, firstly to explore the molecular protective mechanism of VD_3_ and its antioxidant and anti-inflammatory effects against TAA-induced hepatic fibrosis, and secondly to investigate if CD34 and FGF23 could act as novel fibrotic markers and genetically upregulate or downregulate in hepatic tissue in correlation with the incidence and amelioration of liver fibrosis.

## 2. Materials and Methods

### 2.1. Chemicals

Thioacetamide was obtained from Sigma-Aldrich (Munich, Germany), and Devarol S 200000 ampoules (Cholecalciferol) (Memphis Company, Cairo, Egypt) were used for VD_3_ treatment.

### 2.2. Animals and Experimental Design

Thirty-two male albino rats (100–120 g) were obtained from the laboratory animal unit of Zagazig University. During the housing period, rats were fed a standard rodent pellet diet ad libitum with unrestricted access to tap water and were housed in conditions of a 12 h light/dark cycle with consistent temperature and humidity, in compliance with the institutional policies of the Faculty of Veterinary Medicine’s Animal Research Ethical Committee at Mansoura University in Egypt (Ph.D/46).

After 2 weeks of acclimatization, rats were handled carefully and divided into 4 equal groups, each with 8 rats, and kept for 8 consecutive weeks: control group (Cont.) were administered IP injection of normal saline and I/M sterilized distilled water; thioacetamide (TAA) group were administered IP injection of TAA at 100 mg/kg BW every other day according to [22,23]; VD_3_ (VD) group were administered IM injection of Devarol S 200000 at 1000 IU/kg BW every day according to [24]; and dual treatment TAA+VD group were administered the same doses of TAA and VD_3_.

### 2.3. Handling of Biological Samples

An IP injection with a mixture of ketamine and xylazine at a dose of 50 or 10 mg/kg was administered for animal anesthesia at the end of the eighth week of our research. Blood samples were collected from the retro-orbital plexus in plain tubes, then centrifuged at 1500× *g* for 10 min, and serum was collected in Eppendorf tubes and chilled at −20 °C for biochemical analysis. Liver tissues were divided into 3 pieces: the first part, weighing 0.5 g, was crushed in 5 mL ice-cold PBS (PH 7.5) and centrifuged at 2000× *g* for 15 min at 4 °C, and the supernatant was carefully aspirated in Eppendorf tubes and frozen at −20 °C for evaluation of liver oxidative and antioxidant biomarkers; the second portion was preserved in RNAlater for RNA extraction and stored at −80 °C for gene expression analysis; and the last portion was fixed in 10% neutral buffered formalin and prepared for H&E and Masson’s trichrome staining of liver architecture and assessment of fibrosis scoring, in addition to immunohistochemical examination.

### 2.4. Assessment of Serum Biochemistry Parameters

Alanine aminotransferase (ALT), aspartate aminotransferase (AST) (Human Company, Wiesbaden, Germany) and alkaline phosphatase (ALP) (EliTech Company, Paris, France) were quantified spectrophotometrically. Total and direct bilirubin (Diamond Company, Cairo, Egypt) were measured calorimetrically then subtracted to estimate indirect bilirubin. Total protein, albumin, total triglyceride (TG), total cholesterol (TC), and glucose (Spinreact Co., Girona, Spain) were also evaluated. Enzyme-linked immunosorbent assay (ELISA) kits (Diametra Company, Perugia, Italy) were used to estimate serum 25(OH)D_3_ concentration by immunoenzymatic colorimetry [25]. The detection limit for 25(OH)D_3_ assay was 0.3 ng/mL, then repeatability values were acceptable as the intra- and inter-assay coefficients of variability (CV) were ≤6.4 and 6.95%, respectively, and the assay was linear on dilution. A total of 54 serum samples were tested for the comparison of the utilized kits against the commercially available validated kits and yielded an acceptable correlation value (r = 0.829) using the linear regression curve.

### 2.5. Evaluation of Antioxidant and Hepatic Lipid Peroxidation Markers

Hepatic homogenate was utilized to estimate oxidative stress and antioxidant markers as malondialdehyde (MDA) (cat. no. MD 25 29), superoxide dismutase (SOD) (cat. no. SD2521), catalase (CAT) (cat. no. CA 2517), and reduced glutathione (GSH) (cat. no. GR 2511) by commercially available Biodiagnostic (Cairo, Egypt) ready-made kits, which were assessed following the manufacturer’s protocol.

### 2.6. Hepatic Gene Expression Analysis

Using an RNeasy Mini Kit (cat. no. 74104; Qiagen, Hilden, Germany) in accordance with the manufacturer’s instructions, hepatocyte RNA extraction was carried out. Verification of RNA samples was performed using a QuantiTect SYBR Green PCR kit (cat. no. 204141) and a NanoDrop spectrophotometer, including 2 mL of RNase-free water and 1 mL of 2x QuantiTect SYBR Green PCR Master Mix, and Revert Aid Reverse Transcriptase (cat. no. EP044; Thermo Fisher, Waltham, MA USA. Using oligonucleotide primers and probes, SYBR Green real-time PCR was performed to detect fibroblast growth factor 23 (FGF23) and cluster differentiation 34 (CD34) (Metabion, Steinkirchen, Germany). Primer sequences are listed in Table 1. Stratagene Mx3005P software was used to calculate amplification curves and CT values.

According to the CT approach described in [29], the CT of each sample was compared with that of the control group in order to assess the variation of gene expression in the RNA of the samples. β-actin was utilized as a housekeeping reference gene [26].

### 2.7. Liver Histopathology Assessment

According to a previously described technique in [30,31], a formalin-fixed liver specimen was preserved, embedded in paraffin and sectioned into 5 mm thick pieces, and stained with H&E and Masson’s trichrome stain to determine the severity of liver injury. Ishak’s modified Knodell histological activity index (HAI) was used to evaluate the fibrosis score, steatosis grade, and degree of hepatic lobular and portal inflammation.

### 2.8. Immunohistochemistry of α-SMA and TGF-β Expression

The hepatic tissue was cut into 5 µ thick sections from the paraffin blocks and placed on coated slides and deparaffinized, then autoclaved at 120 °C for 10 min in a citrate buffer at pH 6 to retrieve the antigen. The sections were blocked with a 3% H_2_O_2_ solution. The sides were incubated with primary antibodies: rabbit polyclonal TGF-β (1:150) (Wuhan Servicebio Technology Co. Ltd., Wuhan, Hubei, China) and rabbit polyclonal α-SMA (ready to use) (Wuhan Servicebio Technology Co. Ltd., Hubei, China), at room temperature for 1 h. The slides were incubated with peroxidase-conjugated anti-rabbit secondary antibodies at room temperature for 30 min and labeling was “visualized” by incubation with 3,3′ diaminobenzidine tetrahydrochloride as chromogen at room temperature for 5 min [11]. The intensity of positive immunoexpression for TGF- β and α -SMA was scored as 0 = negative, 1 = mild, 2 = moderate, 3 = strong and 4 = very strong in all groups according to [32].

### 2.9. Statistical Analysis

One-way analysis of variance (ANOVA) was used to examine the data as mean ± standard error of the mean, then values were entered into the SPSS software (USA, version 26) and Duncan multiple comparison tests were run afterward. The results were deemed statistically significant at *p* < 0.05 [33].

## 3. Results

### 3.1. Influence of TAA and VD_3_ on Serum Biochemical Parameters

The induction of hepatic damage with TAA resulted in a significant increase (*p* < 0.05) in the serum values of ALT, AST, ALP, total bilirubin and direct bilirubin, and a marked reduction in the levels of total protein and albumin in comparison to the control group. Treatment with VD_3_ significantly restored normal ALT activity in the TAA+VD group compared to the TAA treated group. In addition, AST, ALP, and direct and indirect bilirubin values were downregulated significantly after treatment with TAA+VD compared to the TAA-treated group, but did not return to their normal values (Table 2). Meanwhile, total protein (TP) and albumin (Alb) levels did not significantly change in the TAA+VD group compared to the TAA group (Table 2).

Our results clarify that cholesterol and triglyceride were significantly increased in TAA-treated rats compared to the control group, then both improved after treatment with VD3 but did not return to their normal levels (Table 3). Serum glucose levels were significantly decreased in the TAA group compared to the control, whereas VD3 treatment in the TAA+VD group modified glucose levels (Table 3). Moreover, cholesterol, triglyceride, and glucose were insignificantly affected in VD3 alone with respect to the control group (Table 3). Serum 25[OH] D3 levels were significantly reduced in the TAA group compared to the control. In contrast, they were significantly elevated in rats treated with both VD_3_ and TAA, compared to the control and TAA groups (Table 3).

### 3.2. Influence of TAA and VD_3_ on Hepatic Lipid Peroxidation/Antioxidant Markers

As demonstrated in Figure 1, TAA-treated rats exhibited significant elevation in hepatic MDA levels (95.59 ± 7.5 vs. 60.25 ± 1.55), while hepatic antioxidant biomarkers SOD, CAT, and GSH were substantially decreased compared to the control group (339 ± 46.7 vs. 686 ± 18.6, 1.79 ± 0.04 vs. 1.98 ± 0.01, 4.95 ± 0.33 vs. 7.38 ± 0.23, respectively). Moreover, treatment with VD3 (TAA+VD group) reduced the elevated MDA levels (77.88 ± 6.05 vs. 95.59 ± 7.5) and significantly improved SOD and CAT activity and GSH levels (473 ± 21.72 vs. 339 ± 46.7, 1.91 ± 0.02 vs. 1.79 ± 0.04, 6.09 ± 0.01 vs. 4.95 ± 0.33, respectively), although they did not reach their control levels. Additionally, we did not record any differences between VD3 treated and control rats in oxidative and antioxidant biomarkers.

### 3.3. Influence of TAA and VD_3_ on Hepatic CD34 and FGF23 Gene Expression

Compared to the control group, fibrotic markers CD34 and FGF23 markedly increased in the TAA-treated group (6.65 ± 0.32 vs. 1 ± 0 and 11.21 ± 0.31 vs. 1 ± 0, respectively). On the other hand, these fibrotic markers were significantly reduced in the TAA+VD group compared to the TAA-treated group (2.12 ± 0.15 vs. 6.65 ± 0.32 and 3.98 ± 0.24 vs. 11.21 ± 0.31, respectively). In rats treated with VD3 alone, both fibrotic parameters (CD34 and FGF23) were insignificantly affected with respect to the control group (Figure 2).

### 3.4. Histopathological Examination of Liver Sections Using H&E Staining

Hepatic cords, central veins, portal regions, and sinusoids were all arranged normally in the liver slices from control and VD groups (Figure 3A,B,E,F). The arrangement of hepatic cords, central veins, and portal areas was significantly disrupted in liver sections from the TAA group, which were characterized by thick anastomosing fibrous tissue deposition infiltrated with leukocytes and hemosiderin-laden macrophages containing congested blood vessels and many apoptotic hepatocytes forming multiple distinct complete hepatic nodules. Liver sections from the TAA+VD group showed markedly decreased hepatic lesions, characterized by thin anastomosing fibrous strands extending from portal areas containing few congested blood vessels and very few apoptotic hepatocytes forming few complete hepatic nodules (Figure 3C,D,G,H). Hepatic steatosis grade, lobular inflammation, portal inflammation, and fibrosis score were significantly increased in the TAA group compared to the control group. Meanwhile, treatment with VD3 (TAA+VD) reduced these scores but did not return them to normal levels (Figure 3I–L).

### 3.5. Histopathological Examination of Liver Sections Using Masson’s Trichrome

Liver sections showed no fibrosis in control or VD groups (Figure 4A,B,E,F), whereas the TAA group showed dense bluish-stained fibrous tissue deposition surrounding hepatic nodules. The TAA+VD group showed markedly decreased bluish-stained fibrous tissue deposition (Figure 4C,D,G,H). Fibrosis scoring by Masson’s trichrome was significantly increased in the TAA group compared to the control group. Treatment with VD_3_ (TAA+VD) reduced this score but did not return it to the normal level (Figure 4I).

### 3.6. Immunohistochemical Expression of TGF-β and α-SMA

As explained in Figure 5 and Figure 6, TGF-β and α-SMA expression in the control and VD groups showed negative staining against TGF-β and α-SMA. Meanwhile, the TAA-treated group showed strong positive brown cytoplasmic expression against TGF-β in several hepatocytes, with strong positive brown expression against α-SMA in fibrous tissue and many adjacent hepatocytes. The TAA+VD group showed markedly decreased positive brown cytoplasmic expression against TGF-β in few hepatocytes and α-SMA expression in fibrous tissue and very few adjacent hepatocytes.

## 4. Discussion

Hepatic fibrosis is a reversible multicellular injury that is a healing response to hepatic damage as hepatocytes are replaced by acellular scar tissue, maintaining persistently harmful stimuli, and ECM is heavily deposited [34]. One of the most important hepatotoxicants used extensively in rat models is TAA, as it is extensively metabolized to very reactive metabolites, resulting in hepatic necrosis and oxidative stress [35]. The induction of hepatic damage with TAA was proved in our study, with a significant increase in serum ALT, AST, ALP, total bilirubin, and direct bilirubin activity and a marked reduction in the serum total protein and albumin levels in comparison with the control group, which is in agreement with previous results [22,36]. ROS, which attack lipids, protein, and DNA, usually results in hepatic enzyme leakage of ALT, AST, and ALP in the serum along with hepatocyte damage, centrilobular necrosis, and a substantial reduction in hepatocyte production of total protein and albumin, as well as detoxifying activity [7,11,37].

Modern hepatic fibrosis treatments target fibrogenic myofibroblasts, reduce ECM accumulation, modify fibrosis-relevant pathways, and improve inflammatory signaling in addition to eliminating the underlying cause [34]. VD_3_ significantly lowers the risk of hepatic decompensation by decreasing several inflammatory cytokines in advanced liver fibrosis and cirrhosis, so it has a protective effect on hepatic cells [38]. Interestingly, our serum liver markers were significantly abrogated by treatment with VD_3_; ALT activity returned to normal in the dual injection group compared to the TAA and control groups. AST and ALP activity and total and direct bilirubin were downregulated significantly by treatment with VD_3_ after TAA hepatotoxicity as compared with the TAA-treated group, but did not return to normal values. Meanwhile, indirect bilirubin, TP, Alb and globulin levels and the A/G ratio did not significantly change after dual therapy compared to the TAA and control groups. Our results parallel those of previous studies [38,39] using TAA-induced hepatotoxicity models.

Triglycerides and cholesterol levels after TAA hepatotoxicity showed significantly increased values compared to the control group, which may be attributed to disturbances in the lipid metabolism induced by TAA administration [37]. VD_3_ has lipid-lowering ability, which may explain the improvement in serum cholesterol and triglyceride levels in TAA-intoxicated rats, due to the reduction of gluconeogenesis and lipogenesis through the Ca+2/calmodulin-dependent kinase kinase-β/AMP-activated protein kinase (Ca+2/CaMKK-β/AMPK) pathway, which reduces glucose output and hepatic TG accumulation [40,41]. In this study, the serum of TAA-intoxicated rats showed marked hypoglycemia, as TAA has a temporary impact on the liver’s ability to produce glycogen but later causes the glucose level to change, which disrupts pancreatic β-cells and increases insulin secretion [42,43]. On the other hand, the active form of VD_3_ improved glucose levels in the dual treated group, which was attributed to its ability to elevate insulin response to glucose transport, oxidation in cells, and insulin receptor gene expression (INSR) [44,45].

Oxidative stress is defined as an imbalance between local ROS production and antioxidant mechanisms following hepatic damage [7]. In our work, TAA-induced hepatic damage led to severe oxidative stress, which was shown by a sharp rise in hepatic MDA levels and a significant decrease in the levels of endogenous intracellular antioxidant enzymes CAT, SOD, and GSH in liver homogenate. TAA is metabolized by two-step bioactivation of the flavin-containing monooxygenase (FMO) system and cytochrome P450 (Cyt-p450) enzyme present in the liver microsomes, called thioacetamide S-oxide (T-ASO), reducing dioxygen to superoxide anion, which is catalyzed to form hydrogen peroxide [22,43]. Rats administered TAA were found to exhibit a significant elevation in liver peroxidation marker levels with a consequent reduction in antioxidant parameter levels, compared to control rats in many previous studies with different TAA doses and durations [7,39,43,46,47].

The antioxidant capacity of VD_3_ against ROS production from a damaged liver was determined in our study by the inhibition of lipid peroxidation due to decreased MDA levels in hepatic homogenates. Treatment with VD_3_ significantly improved SOD and CAT activity and GSH levels compared to TAA treatment, although they did not reach the control levels. Additionally, we did not record any difference between VD_3_-treated and control rats in oxidative and antioxidant biomarkers. The protective action of VD_3_ was proven by decreased endotoxemia in endothelial cells and their apoptosis, which reduces the formation of free radicals and lipid peroxidation in liver disorders [38,48]. VD_3_ has dual action; it controls free radical generation and reduces oxidative stress, either through binding to the vitamin D receptor (VDR) in the nucleus or through the hydrophobic parts of VD, depending on the type of cell, even in cells that lack a nucleus such as mature erythrocytes [49].

The fibrogenic response comprises four major stages: (1) the primary cause of liver injury triggers and initiates a fibrosis response, (2) effector cells are activated, (3) the extracellular matrix is elaborated, and (4) there is a dynamic and progressive deposition of ECM leading to liver failure, with interactions among many complex and dynamic mechanisms of resident cellular architecture in hepatocytes, resident/non-resident progenitor cells, locally released growth factors and cytokines, and systemically released chemokines [50].

Compared with the control group in our study, fibrotic markers CD34 and FGF23 markedly increased in the TAA-treated group and were significantly reduced after VD_3_ treatment. CD34 was identified as a fibrotic marker as tissue regeneration in subcutaneous wounds was associated with fibrocytes, a circulating bone marrow-derived CD34+ cell subset with fibroblast-like characteristics [51]. Fibrogenic cell populations were produced from bone marrow in the CCl4 fibro hepatic mouse and the bile duct ligation model of biliary hyperplasia [52,53]. In the Abcb4_/_ mice, well-characterized non-surgical models for cholangiopathy in humans, bone marrow-derived circulating CD34+ fibrocytes were proven to serve as essential mediators of liver fibrogenesis [54]. Hepatic fibrosis, hepatocyte regeneration, and the migration, proliferation, and transdifferentiation of HSCs are all influenced by FGFs. Numerous different FGF subfamilies have been demonstrated to influence fibrogenesis. FGF23 is also expressed in damaged liver tissue, although it is only very weakly expressed in healthy liver and, on the same line, FGF23 has been shown to promote cardiac and renal fibrosis [55,56]. The TAA-induced liver injury greatly boosted FGF23 expression in diethylnitrosamine-induced liver injury in mice [57]. On the same line, FGF23 has been shown to promote cardiac and renal fibrosis.

The arrangement of hepatic cords, central veins, and portal areas with H&E staining was significantly disrupted in liver sections, with many apoptotic hepatocytes forming multiple distinct complete hepatic nodules; moreover, Masson’s trichrome staining showed dense bluish-stained fibrous tissue deposition surrounding hepatic nodules. The alleviating and antifibrotic effects of VD_3_ on the hepatic parenchyma after TAA liver fibrosis was confirmed in our research by the presence of very few apoptotic hepatocytes forming few complete hepatic nodules; furthermore, Masson’s trichrome staining showed markedly decreased bluish-stained fibrous tissue deposition. Our staining images were also in agreement with the results reported in [46,58], in which they monitored the changes in hepatic fibrosis and necroinflammation in male rats that received IP TAA injections at 150 mg/kg twice weekly for twelve weeks and 100 mg/kg three times weekly for eight weeks, respectively.

Hepatic steatosis, lobular inflammation, portal inflammation, and fibrosis scores by H&E and Masson’s trichrome staining were significantly increased in the TAA group compared to the control group. Treatment with VD_3_ after TAA-induced hepatic fibrosis resulted in decreased scores, but the scores did not return to normal levels [31,58].

Two essential players in the growth of liver fibrosis are TGF-β1 and α -SMA, as they are pleiotropic inflammatory cytokines known to be markers of HSC activation that transform into myofibroblasts and then further enhance the transcription of collagen and other ECM components [59,60]. Negative staining of the TGF-β and α-SMA immunohistochemical expression in hepatic tissue was evident in our control and VD groups. A strong positive brown cytoplasmic expression was seen against TGF-β and α-SMA in several hepatocytes, fibrous tissue, and many adjacent hepatocytes in our fibrotic TAA model. VD_3_ injection after TAA-induced fibrosis showed markedly decreased positive brown cytoplasmic expression against TGF-β and α-SMA in fibrous tissue and very few adjacent hepatocytes. Our findings concur with previous studies on the antifibrotic effect of VD_3_ [61]. In addition, treating rats with VD_3_ alone inhibited TGF-β-induced fibroblast proliferation and α -SMA expression [62], as VD_3_ exerts its protective effect on the progression of liver fibrosis via an antifibrotic effect on HSCs, regulating cell proliferation and differentiation, and modulating a specific signal transduction pathway mediated by VD receptors [63].

## 5. Conclusions

VD_3_ acts as a hepatoprotective agent against TAA-induced hepatotoxicity and fibrosis by stabilizing the antioxidant system with oxidative inflammatory status as well as modulating the TGF-β and α-SMA fibrotic molecule pathways.

## Figures and Tables

**Figure 1 biomedicines-11-00424-f001:**
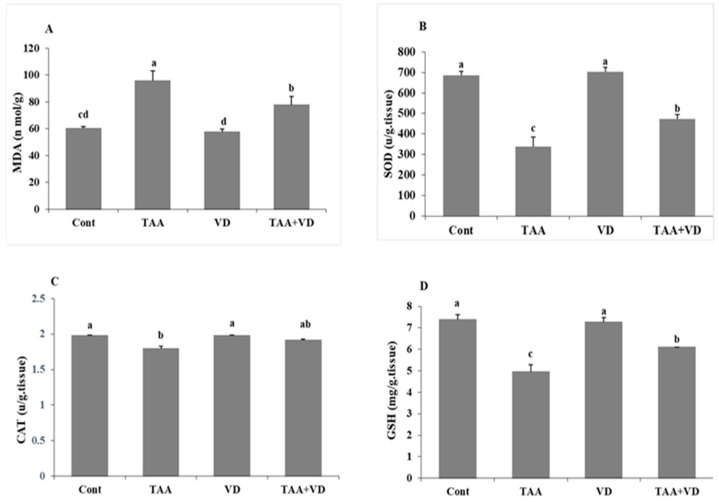
Hepatic oxidative/antioxidative parameters at the end of the eighth week post treatment with VD_3_ against TAA-induced hepatic damage in rats (mean ± SE). Various superscripts (a, b, c, and d) are statistically different from one another (*p* < 0.05): (**A**) MDA (nmol/g tissue), (**B**) SOD (U/g tissue), (**C**) GSH (mg/g tissue), and (**D**) catalase (U/g tissue). MDA, malondialdehyde; SOD, superoxide dismutase; CAT, catalase; GSH, reduced glutathione.

**Figure 2 biomedicines-11-00424-f002:**
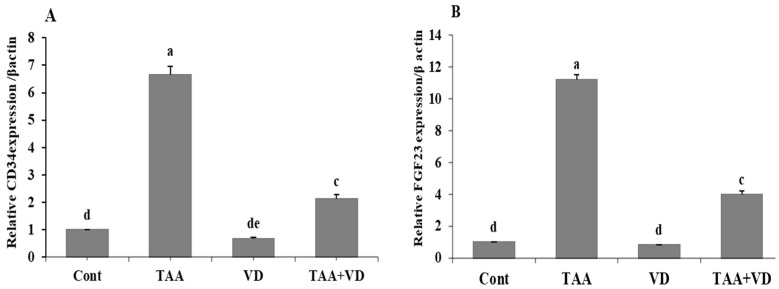
(**A**) CD34 and (**B**) FGF23 in different experimental groups at the end of the eighth week post treatment with VD3 against TAA-induced hepatic damage in rats (mean ± SE). Various superscripts (a, c, d, and e) are statistically different from one another (*p* < 0.05). CD34, cluster differentiation 34; FGF23, fibroblast growth factor 23.

**Figure 3 biomedicines-11-00424-f003:**
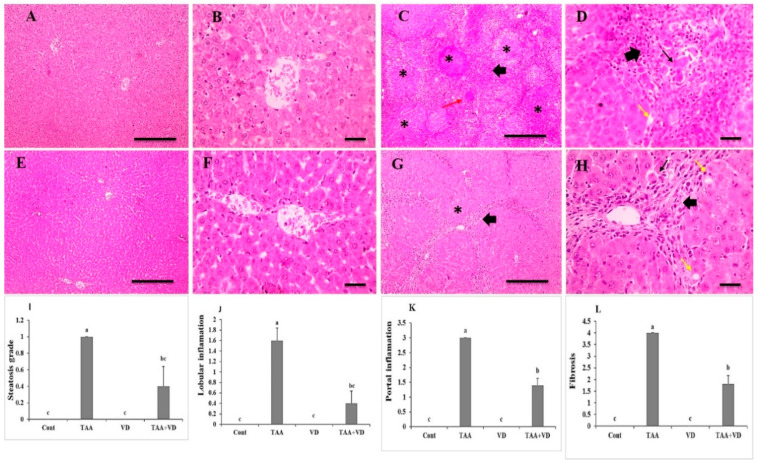
Histopathological examination of H&E-stained liver sections showing (**A**,**B**) normal arrangement of hepatic cords, central veins, portal areas, sinusoids in control group and (**C**,**D**) markedly disrupted arrangement of hepatic cords, central veins, and portal areas, characterized by thick anastomosing fibrous tissue deposition infiltrated with leukocytes and hemosiderin-laden macrophages (thick black arrows) containing congested blood vessels (red arrows) and many apoptotic hepatocytes (thin arrows) forming separate complete hepatic nodules (*) in TAA group. (**E**,**F**) Normal arrangement of hepatic cords, central veins, portal areas, sinusoids in VD group. (**G**,**H**) Markedly decreased hepatic lesions, characterized by thin anastomosing fibrous strands extending from portal areas (thick black arrows) containing few dilated blood vessels (red arrows) and very few apoptotic hepatocytes (thin arrows) forming incomplete hepatic nodules in TAA+VD group. Magnification (µm): 100 bar = 100×; 400 bar = 50×. (**I**) Hepatic steatosis, (**J**) hepatic lobular inflammation, (**K**) hepatic portal inflammation, and (**L**) hepatic fibrosis grades in different experimental groups. Each bar represents mean ± SEM (n = 8). Bars with different superscript letters are significantly different (*p* < 0.05). Cont., control; TAA, thioacetamide; VD, VD_3_; TAA+VD, thioacetamide + VD_3_.

**Figure 4 biomedicines-11-00424-f004:**
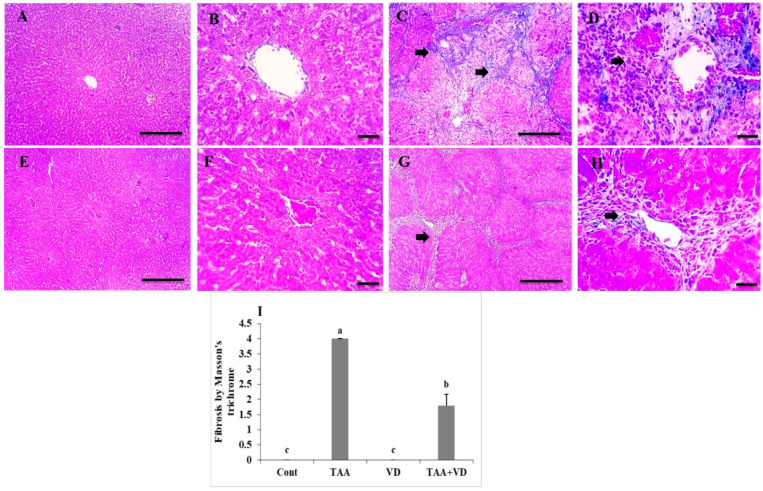
Histopathological examination of Masson’s trichrome stained liver sections. (**A**,**B**) Absence of fibrous connective tissue in control group. (**C**,**D**) Dense bluish-stained fibrous tissue deposition surrounding hepatic nodules (black arrows) in TAA-treated group. (**E**,**F**) Absence of fibrosis in VD group. (**G**,**H**) Marked decrease in thickness of deposited bluish-stained fibrous tissue (black arrows) in TAA+VD group. Magnifications (µm): 100 bar = 100×; 400 bar = 50×. (**I**) Hepatic fibrosis scoring by Masson’s trichrome in experimental groups; each bar represents mean ± SEM (n = 8). Bars with different superscript letters are significantly different (*p* < 0.05). Cont., control; TAA, thioacetamide; VD, VD_3_; TAA+VD, thioacetamide + VD_3_.

**Figure 5 biomedicines-11-00424-f005:**
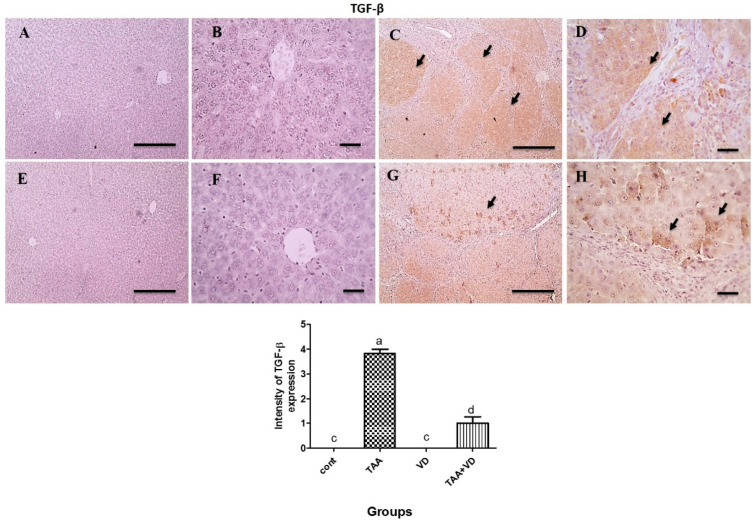
Microscopic pictures of immune stained liver sections. (**A**,**B**) Negative staining against TGF-β in control group. (**C**,**D**) Strong positive brown cytoplasmic expression against TGF-β in several hepatocytes (black arrows) in TAA-treated group. (**E**,**F**) Negative staining against TGF-β in VD group. (**G**,**H**) Markedly decreased positive brown cytoplasmic expression against TGF-β in few hepatocytes (black arrows) in TAA+VD group. IHC counterstained with Mayer’s hematoxylin. Magnification (µm): 100 bar = 100×; 400 bar = 50×. Positive immunoexpression for TGF- β in the different experimental groups, each bar represents mean ± SEM (n = 8). Bars with different superscript letters are significantly different (*p* < 0.05). Cont., control; TAA, thioacetamide; VD, VD_3_; TAA+VD, thioacetamide + VD_3_.

**Figure 6 biomedicines-11-00424-f006:**
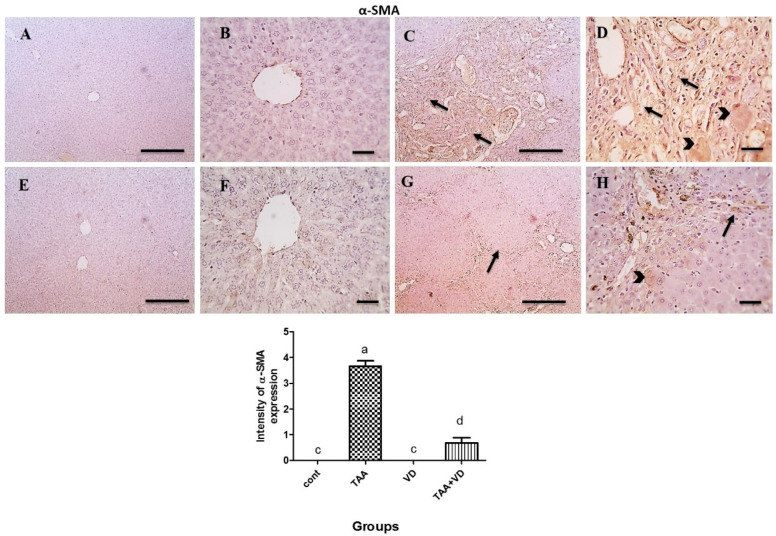
Microscopic pictures of immune stained liver sections. (**A**,**B**) Negative staining against α-SMA in control group. (**C**,**D**) Strong positive brown expression against α-SMA in fibrous tissue (black arrows) and adjacent hepatocytes adjacent hepatocyte (Black arrow heads) in TAA treated group. (**E**,**F**) Negative staining against α-SMA in VD group (**E**,**F**) Negative staining against α-SMA in VD group. (**G**,**H**) Markedly decreased positive brown cytoplasmic expression against α-SMA expression in fibrous tissue (Black arrows) and very few adjacent hepatocytes (Black arrow heads) in TAA+VD group. IHC counterstained with Mayer’s hematoxylin. Magnification (µm): 100 bar = 100×; 400 bar = 50×. Positive immunoexpression for α-SMA in the different experimental groups, each bar represents mean ± SEM (n = 8). Bars with different superscript letters are significantly different (*p* < 0.05). Cont., control; TAA, thioacetamide; VD, VD_3_; TAA+VD, thioacetamide + VD_3_.

**Table 1 biomedicines-11-00424-t001:** Primer sequences for RT-PCR.

Gene	Primer Sequence (5′-3′)	Reference
Rat β-actin	F:3′TCCTCCTGAGCGCAAGTACTCT5′ R:5′GCTCAGTAACAGTCCGCCTAGAA3′	[26]
CD34	F:3′AGCCATGTGCTCACACATCA5′ R:5′CAAACACTCGGGCCTAACCT3′	[27]
FGF23	F:3′ACGGAACACCCCATCAGACTATC5′ R: 5′TATCACTACGGAGCCAGCATCCTC3′	[28]

**Table 2 biomedicines-11-00424-t002:** Serum liver function indicators at the end of the eighth week post-treatment with VD_3_ against TAA-induced hepatic damage in rats (mean ± SE).

Parameter	Cont.	TAA	VD	TAA+VD
ALT (U/L)	43.50 ± 1.17 ^b^	362.38 ± 47.56 ^a^	42.94 ± 0.75 ^b^	103.56 ± 2.78 ^b^
AST (U/L)	136.80 ± 10.32 ^c^	1178.20 ± 24.58 ^a^	130.8 ± 12.06 ^c^	600 ± 154.36 ^b^
ALP (U/L)	148.20 ±8.95 ^c^	424.80 ± 49.72 ^ab^	127.70 ± 7.27 ^c^	330.20 ± 11.52 ^b^
Total bilirubin (mg/dL)	0.31 ± 0.05 ^c^	1.28 ± 0.18 ^a^	0.41 ± 0.07 ^c^	0.89 ± 0.06 ^b^
Direct bilirubin (mg/dL)	0.12 ± 0.03 ^c^	0.95 ± 0.18 ^a^	0.19 ± 0.04 ^c^	0.62 ± 0.08 ^b^
Indirect bilirubin (mg/dL)	0.18 ± 0.04 ^a^	0.33 ± 0.00 ^a^	0.21 ± 0.05 ^a^	0.27 ± 0.06 ^a^
Total protein (g/dL)	8.42 ± 0.20 ^a^	6.82 ± 0.24 ^b^	8.13 ± 0.24 ^a^	6.99 ± 0.25 ^b^
Albumin (g/dL)	4.13 ± 0.14 ^a^	3.09 ± 0.11 ^b^	4.06 ± 0.13 ^a^	3.38 ± 0.15 ^b^
Globulin (g/dL)	4.29 ± 0.21 ^a^	3.73 ± 0.24 ^a^	4.07 ± 0.18 ^a^	3.61 ± 0.19 ^a^
A/G ratio	0.97 ± 0.06 ^a^	0.84 ± 0.06 ^a^	1.00 ± 0.05 ^a^	0.94 ± 0.05 ^a^

Values are presented as mean ± SEM. Those containing various superscripts (a, b, and c) are statistically different from one another (*p* < 0.05). Cont., control; TAA, thioacetamide; VD, VD_3_; TAA+VD, thioacetamide + VD_3_; ALT, alanine aminotransferase; AST, aspartate aminotransferase; ALP, alkaline phosphatase; A/G, albumin/globulin.

**Table 3 biomedicines-11-00424-t003:** Some serum biochemical markers at the end of the eighth week post-treatment with VD_3_ against TAA-induced hepatic damage in rats (mean ± SE).

Parameter	Cont.	TAA	VD	TAA+VD
TC (mg/dL)	50.49 ± 2.29 ^b^	67.64 ± 1.44 ^a^	53.57 ± 0.71 ^b^	58.55 ± 1.19 ^ab^
TG (mg/dL)	100.26 ± 4.06 ^c^	165.80 ± 10.96 ^a^	99.36 ± 6.48 ^c^	133.40 ± 11.66 ^b^
Glucose (mg/dL)	96.26 ± 2.88 ^a^	68.13 ± 4.60 ^b^	94.60 ± 3.56 ^a^	89.25 ± 8.6 ^ab^
VD (25[OH] D_3_) (ng/mL)	34.35 ± 2.05 ^c^	2.76 ± 0.14 ^d^	148.33±4.40 ^a^	122.33±1.45 ^b^

Values are presented as mean ± SEM. Those containing various superscripts (a, b, c, and d) are statistically different from one another (*p* < 0.05). Cont., control; TAA, thioacetamide; VD, VD_3_; TAA+VD, thioacetamide + VD_3_; TC, total cholesterol; TG, triglyceride.

## Data Availability

Data available within the manuscript.

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
