# Peer review of "Vitamin D Ameliorates the Hepatic Oxidative Damage and Fibrotic Effect Caused by Thioacetamide in Rats"

_biomedicines, 2023, doi:10.3390/biomedicines11020424_

Round 1
Reviewer 1 Report
The study was well done and might be interesting, but there were the points to be reconsidered.
1. General impression: there have been many papers indicating the positive associations of vitamin D3 and liver pathology including fibrosis. The authors can stress more the novelty and relevance of the study on this topic field for many readers.
2. Introduction; although the authors described that the animal model resembles human liver pathology (as ref. 9), to what extent/degree does the model resemble it (highly or moderately)?
3. Methods; Although the authors measured CD34 and FGF23 as fibrotic markers, the authors can explain the validity to select those markers.
4. Methods; Although the authors measured some oxidative and anti-oxidative markers, why did the authors select such markers? For instance, other markers including lipid peroxidation are candidates for the study.
5. Methods; The measurement of Vitamin D (25[OH]D3) is currently known not to be standardized. The assay validation can be described concretely.
6. Methods; The statistical significance should be more clearly stated in one sentence.
7. Discussion; Although the authors described the application of the study findings to human settings in the last parts, are there any human data like the present study’s data?
8. Discussion; Although the authors described the misconception of CD34 on cells, many readers may not immediately understand it. The meaning of sentences may be more explained for many readers.
9. Overall text; there were several parts showing inconsistency of words of vitamin D3 as ‘Vit-D’, ‘vitamin D3’ ‘VD3’, VD3’ etc.
10. Overall text and Table; there were several parts showing a typo in the non-beginning of sentence as ‘On The other..’, ‘Alkaline…’, ‘Total Bilirubin’ etc.
11. Text; the abbreviation should be used for later and, for example, be rechecked in MSC as seen in Discussion section.
12. Text; the space should be consistently required between sentence and [ref no.]. Recheck the overall text.
13. Text; the space should be deleted before ‘Therefore,’ in the last sentence in Introduction section.
14. English check (i.e., the aspects of grammar and expression) is required to be readable more easily.
Author Response
Response for reviewer 1:
Thank you for your constructive comments concerning our manuscript. We have studied your comments carefully and made major correction which we really hope to meet with your approval.
Comments and Suggestions for Authors
The study was well done and might be interesting, but there were the points to be reconsidered.
- General impression: there have been many papers indicating the positive associations of vitamin D3 and liver pathology including fibrosis. The authors can stress more the novelty and relevance of the study on this topic field for many readers.
Response: The main objective of the present study was to address the ability of vitamin D3 treatment to ameliorate TAA-induced histologically confirmed hepatic fibrosis through the estimation of liver function and hepatic oxidative stress markers and antioxidant biomarkers in combination with the gene expression of fibrotic markers in rats. To the author knowledge, our study considers the first experiment that revealed the incidental upregulation of novel fibrotic markers as CD34 and FGF23 expression as well as traditional fibrotic markers, α -SMA and TGF-β in correlation with the incidence of hepatic fibrosis and their downregulation in ameliorated hepatic fibrosis. These findings could open the way for more understanding of the hepatic fibrosis that will help in the future finding of the effective therapeutic intervention through down regulation of mentioned fibrotic markers pathway. Moreover, estimated genes (CD34 and FGF23) can act as a new reflective molecular biomarker of hepatic fibrosis initiation in the field of molecular diagnosis, that confirmed by using Masson’s trichrome to hepatic tissue and immunohistochemical expressions of fibrotic markers α-SMA and TGF-β. The novelty and relevance of the study is collective these parameters in one study has been mentioned at the end of the introduction section in the manuscript.
2-Introduction; although the authors described that the animal model resembles human liver pathology (as ref. 9), to what extent/degree does the model resemble it (highly or moderately)?
Response:
As mentioned by An et al. (2006); Belanger et al. (2005) and Stankoya et al. (2010) thioacetamide administration either intraperitoneally or in drinking water up to 6 months produced liver cirrhosis and hepatic neoplasms in various animal models with significant biochemical and histological alterations that have shown high similarity to human liver failure.
As mentioned by Yang et al. (1998) TAA induced inflammation leads to cirrhotic conditions in the rat liver that resembles the human alcoholic liver fibrosis. It is a model hepatotoxicant and responsible of severe damage to the cells with significant toxic effects on biosynthesis of macromolecules including proteins and nucleic acids
- An, J.H., Seong, J., Oh, H., Kim, W., Han, K.H., Paik, Y.H. “Protein Expression Profiles in a Rat Cirrhotic Model Induced by Thioacetamide,” Korean Journal of Hepatology, Vol. 12, No. 1, 2006, pp. 93-102.
- Belanger M., Butterworth, R. “Acute Liver Failure: A Critical Appraisal of Available Animal Models,” Metabolic Brain Disease, Vol. 20, No. 4, 2005, pp. 409-423.
- Stankova, P., Kucera, O., Lotkova, H., Rousar, T., Endlicher, R., Cervinkova, Z. “The Toxic Effect of Thioacetamide on Rat Liver in Vitro,” Toxicology in Vitro, Vol. 24, No. 8, 2010, pp. 2097-2103.
- Yang, J.M., Han, D.W., Xie, C.M., Liang, Q.C., Zhao, Y.C., Ma, X.H. “Endotoxins Enhance Hepatocarcinogenesis Induced by Oral Intake of Thioacetamide in Rats,” World Journal of Gastroenterology, Vol. 4, No. 2, 1998, pp. 128-132.
- Methods; Although the authors measured CD34 and FGF23 as fibrotic markers, the authors can explain the validity to select those markers.
Response: In our study CD34 was measured as fibrotic marker according to previous study applied by Roderfeld et al.(2013) who explained that bone marrow-derived circulating CD34+ fibrocytes could serve as essential early mediators of liver fibrogenesis.in the current study FGF23 was significantly expressed in injured liver tissue in comparing with hepatic tissue obtained from the normal control group. Furthers, we noted that previous studies (Prié et al.,2013, Norikazu et al., 2022) revealed that FGF23 was significantly increased in diethylnitrosamine-induced liver injury in mice so we evaluated its expression level in TAA-induced liver fibrosis in attempt to confirm the finding in our fibrotic model.
Roderfeld, M., Rath, T., Voswinckel, R., Dierkes, C., Dietrich H., Zahner, D., Graf, J., Roeb, E. Bone marrow transplantation demonstrates medullar origin of CD34(+) fibrocytes and ameliorates hepatic fibrosis in abcb4(_/_) mice, Hepatology 51 (1) (2010) 267–276.
- Norikazu Toi, Yasuo Imanishi, Yuki Nagata , Masafumi Kurajoh , Tomoaki Morioka , Tetsuo Shoji , Yoshitaka Shinto , Masanori Emoto . I mprovements in the mobility of a patient with fibroblast growth factor 23-related hypophosphatemia osteomalacia and decompensated liver cirrhosis in response to burosumab: a case report. Endocr J2022 Dec 28. doi: 10.1507/endocrj.EJ22-0520.
- Prié, D., Forand, A., Francoz, C., Elie, C., Cohen, I., Courbebaisse, M., ... & Friedlander, G. (2013). Plasma fibroblast growth factor 23 concentration is increased and predicts mortality in patients on the liver-transplant waiting list. PloS one, 8(6), e66182.
- Methods; Although the authors measured some oxidative and anti-oxidative markers, why did the authors select such markers? For instance, other markers including lipid peroxidation are candidates for the study.
Response: There are many recent research papers adopted estimation of liver MDA as a marker for lipid peroxidation process in liver, as well as the SOD, CAT, and GSH have a crucial role in the antioxidant system in the hepatic tissues. Catalase has a key role to prevent cellular oxidative damage by degrading hydrogen peroxide (H 2 O 2) into water and oxygen (free radical) with high efficiency to GSH. When GSH bonds with a free radical, GSH transforms into the oxidized form of glutathione GSSG. The SOD enzymes catalyze the conversion of superoxide, which is a reactive oxygen species that’s produced during aerobic respiration, that transfers energy to cells. [(El-Mihi et al., 2017), (Özdemir-Kumral et al., 2019), (Ghanim et al., 2021), (Mohsen et al., 2021), (Al-Medhtiy et al.,2022)].
- El-Mihi, K. A., Kenawy, H. I., El-Karef, A., Elsherbiny, N. M., Eissa, L. A. (2017). Naringin attenuates thioacetamide-induced liver fibrosis in rats through modulation of the PI3K/Akt pathway. Life sciences, 187, 50-57.
- Özdemir-Kumral, Z. N., Erkek, B. E., Karakuş, B., Almacı, M., Fathi, R., Yüksel, M., Cumbul, A., Alican, İ. (2019). Potential effect of 1, 25 Dihydroxyvitamin D3 on thioacetamide-induced hepatotoxicity in rats. Journal of Surgical Research, 243, 165-172.
- Al-Medhtiy, M. H., Jabbar, A. A., Shareef, S. H., Ibrahim, I. A. A., Alzahrani, A. R., Abdulla, M. A. (2022). Histopathological evaluation of Annona muricata in TAA-induced liver injury in rats. Processes, 10(8), 1613.
- Ghanim, A. M., Younis, N. S., Metwaly, H. A. (2021). Vanillin augments liver regeneration effectively in Thioacetamide induced liver fibrosis rat model. Life Sciences, 286, 120036.
- Mohsen, A. M., Younis, M. M., Salama, A., Darwish, A. B. (2021). Cubosomes as a potential oral drug delivery system for enhancing the hepatoprotective effect of coenzyme Q10. Journal of Pharmaceutical Sciences, 110(7), 2677-2686.
- Methods; The measurement of Vitamin D (25[OH]D3) is currently known not to be standardized. The assay validation can be described concretely.
Response: As mentioned in submitted manuscript vitamin D was evaluated using commercially available kits of Diametra company as highlighted (yellow) in attached pamphlet, the assay validation process such as (intra-assay and inter-assay Precision, sensitivity, and correlation). Beside the Vitamin D was standardized with 6 calibrators are human serum. As well as 2 controls sample (high and low control values) as display in the attached pamphlet.
6.Methods; The statistical significance should be more clearly stated in one sentence.
Response: thank you for your advice, we clarify (Done).
7.Discussion; Although the authors described the application of the study findings to human settings in the last parts, are there any human data like the present study’s data?
Response: Our finding in TAA hepatotoxicant rat model is completely resemble the result obtained by An et al. (2006), Belanger et al. (2005) and Stankoya et al. (2010) in fibrotic liver in human patient.
- Discussion; Although the authors described the misconception of CD34 on cells, many readers may not immediately understand it. The meaning of sentences may be more explained for many readers. Response: Done.
- Overall text; there were several parts showing inconsistency of words of vitamin D3as ‘Vit-D’, ‘vitamin D3’ ‘VD3’, VD3’ etc. Done. Response: Done.
- Overall text and Table; there were several parts showing a typo in the non-beginning of sentence as ‘On The other..’, ‘Alkaline…’, ‘Total Bilirubin’ etc. Response: Done.
- Text; the abbreviation should be used for later and, for example, be rechecked in MSC as seen in Discussion section. Response: Done.
- Text; the space should be consistently required between sentence and [ref no.]. Recheck the overall text. Response: Checked.
- Text; the space should be deleted before ‘Therefore,’ in the last sentence in Introduction section. Response: Done.
- English check (i.e., the aspects of grammar and expression) are required to be readable more easily. Response: our manuscript has been English edited by biomedicine editing services and we believe that now is in a good shape. (Already we have now a certificate which will be attached to the file).

Reviewer 2 Report
Aya Megahed et al. describe a series of experiments where they evaluated the effects of Vitamin D on oxidative damage and in the liver section showed the reduction of fibrotic characteristics such as collagen, TGFb, and aSMA…
All the methodology and the evidence shown here are very interesting and relevant, however, there are some major and minor points corrections
Major
Figure 3 is missing
Minor
Abstract
Homogenize terms such as VD3 or VD.
Conclusions are missing in the Abstract.
Introduction
Check for syntax, the appearance of abbreviations, and in the last section of the introduction is a larger font size
Materials and methods
Authors should provide the number of approvals by the Faculty of Veterinary Medicine’s Animal Research Ethical Committee at Mansoura University in Egypt.
Authors should briefly describe the section
2.5. Evaluation of the antioxidant and hepatic lipid peroxidation markers
Results
The authors should provide in the text the values and significance.
In the tables meaning superindices a, b, c, ab
Figure 3 is missing
In figure 4, the graph does not show a good resolution.
It is a little difficult to follow the arrangement of the images for a better understanding, fig 4 should be rearranged.
The same with fig 5
It would be interesting to see the quantification of tgfb and asma.
Author Response
Response for reviewer 2:
We would like to thank the reviewer for detailed comments, careful and thoroughly reading of this manuscript, and for the constructive suggestions, which help to improve the quality of this manuscript.
Aya Megahed et al. describe a series of experiments where they evaluated the effects of Vitamin D on oxidative damage and in the liver section showed the reduction of fibrotic characteristics such as collagen, TGF-βand α-SMA.
All the methodology and the evidence shown here are very interesting and relevant, however, there are some major and minor points corrections
Major
Figure 3 is missing. Response: we apologize about this point and we added in the manuscript (Done).
Minor
1.Abstract:
-Homogenize terms such as VD3 or VD. Response: Done.
-Conclusions are missing in the Abstract. Response: we added a sentence at the end of the abstract (Done).
2.Introduction:
-Check for syntax, the appearance of abbreviations, and in the last section of the introduction is a larger font size. Response: Done.
3.Materials and methods:
-Authors should provide the number of approvals by the Faculty of Veterinary Medicine’s Animal Research Ethical Committee at Mansoura University in Egypt.
Response: We already added in the manuscript (section 2.2 Animals and experimental design Line 4) (in compliance with the institutional policies of the Faculty of Veterinary Medicine’s Animal Research Ethical Committee at Mansoura University in Egypt (Ph.D/46).
-Authors should briefly describe the section
- Evaluation of the antioxidant and hepatic lipid peroxidation markers
Response: We already added in the manuscript (section 2.3 Handling of biological samples Line 5 and section 2.5 Evaluation of antioxidant and hepatic lipid peroxidation markers).
Liver tissues were divided into three pieces; the first part weighing (0.5 g) was crushed in 5 ml ice-cold PBS (PH 7.5) and centrifuged at 2000 gx for 15 minutes at 4°C, supernatant was carefully aspirated in Eppendorf tubes and were frozen at -20 °C for evaluation of liver oxidative and antioxidant biomarkers. Hepatic homogenate was utilized to estimate oxidative stress and antioxidant marker as malondialdehyde (MDA) [Cat No: MD 25 29], superoxide dismutase (SOD) [Cat No: SD2521], catalase (CAT) [Cat No: CA 2517] and reduced glutathione (GSH) [Cat No: GR 2511] by commercially available Bio-Diagnostic ready-made kits [Cairo, Egypt], and were assessed following manufacturer’s protocol.
4-Results:
-The authors should provide in the text the values and significance. Response: We added the values in the manuscript (Done).
-In the tables meaning super indices a, b, c, ab Response: We added a sentence under the tables (Done).
-Figure 3 is missing. Response: we apologize about this point and we added in the manuscript (Done).
-In figure 4, the graph does not show a good resolution. Response: We changed our figures in this section; we wish it will be suitable now (Done).
-It is a little difficult to follow the arrangement of the images for a better understanding, fig 4 should be rearranged. Response: We rearranged according to the reviewer advice; we wish it will be suitable now.
-The same with fig 5. Response: Done.
-It would be interesting to see the quantification of TGF-βand α-SMA. Response: we added in our manuscript figures (5,6 respectively).

Reviewer 3 Report
This original article arouses interest for readers and provides an important clue to treat oxidative stress-induced acute liver injury with vitamin D and discuss the underlying mechanisms by which vitamin D act on TAA-induced liver injury. However, there are some problems that should be addressed or altered.
1) There are too many, many typographic and grammatical errors in this text. For instance, “concussive” in the section 2.2 is correct? Authors should brush up their manuscript more, more and more carefully!
2) Could you provide the sequences of forward and reverse primers and probes in Table 1?
3) What is “HESPERIDIN” in the Figure 2 legend?
4) Pictures in Figure 4 are unclear and thus should be replaced by clear others. For instance, the color of alpha characters had better be white.
5) As described in the Discussion section, FRFR4 is the predominant isoform of the FGFRs in hepatocytes and may have a major role in progression of liver fibrosis. Where is a description of “FGF23”? In addition, why did authors select CD34 and FGF23 as fibrosis markers?
Author Response
Response for reviewer 3:
We appreciate the reviewer's thoughtful observations, diligent and in-depth reading of this manuscript, and helpful suggestions that helped to elevate the work's value.
Comments and Suggestions for Authors
This original article arouses interest for readers and provides an important clue to treat oxidative stress-induced acute liver injury with vitamin D and discuss the underlying mechanisms by which vitamin D act on TAA-induced liver injury. However, there are some problems that should be addressed or altered.
- There are too many, many typographic and grammatical errors in this text. For instance, “concussive” in the section 2.2 is correct? Authors should brush up their manuscript more, more and more carefully! Response: yes, the typographic and grammatical has been Checked carefully throughout the manuscript.
2) Could you provide the sequences of forward and reverse primers and probes in Table 1? Response: thank you for your advice (We added to the table 1).
3) What is “HESPERIDIN” in the Figure 2 legend? Response: thank you for your comments (We deleted from the figure).
4) Pictures in Figure 4 are unclear and thus should be replaced by clear others. For instance, the color of alpha characters had better be white. Response: we replaced this figure by other more cleared one.
5) As described in the Discussion section, FRFR4 is the predominant isoform of the FGFRs in hepatocytes and may have a major role in progression of liver fibrosis. Where is a description of “FGF23”? In addition, why did authors select CD34 and FGF23 as fibrosis markers?
The fibrogenic response comprises four major stages: (1) the primary cause of liver injury triggers and initiates a fibrosis response, (2) effector cells are activated, (3) extracellular matrix is elaborated, and (4) there is dynamic and progressive deposition of ECM leading to liver failure, with interactions among many complex and dynamic mechanisms of resident cellular architecture in hepatocytes, resident/non-resident progenitor cells, locally released growth factors and cytokines, and systemically released chemokines [50]. Compared with the control group in our study, fibrotic markers CD34 and FGF23 markedly increased in the TAA-treated group, and were significantly reduced after VD3 treatment. CD34 identified as fibrotic marker as tissue regeneration in subcutaneous wounds was associated with fibrocytes, a circulating bone marrow-derived CD34+ cell subset with fibroblast-like characteristics [51]. Production of fibrogenic cell populations from bone marrow in the CCl4 fibro hepatic mouse and the bile duct ligation model of biliary hyperplasia [52, 53]. In the Abcb4_/_ mice, well-characterized non-surgical model for cholangiopathy in humans, bone marrow-derived circulating CD34+ fibrocytes were proven to serve as essential mediators of liver fibrogenesis [54]. Hepatic fibrosis, hepatocyte regeneration, and the migration, proliferation, and transdifferentiation of HSCs are all influenced by FGFs. Numerous different FGF subfamilies have been demonstrated to influence fibrogenesis. FGF23 is also expressed in damaged liver tissue, although it is only very weakly expressed in healthy liver and on the same line, FGF23 has been shown to promote cardiac and renal fibrosis [55,56]. The TAA-induced liver injury greatly boosted FGF23 expression in diethylnitrosamine-induced liver injury in mice [57] On the same line, FGF23 has been shown to promote cardiac and renal fibrosis.

Round 2
Reviewer 1 Report
Most parts were improved. The assay precision and accuracy (e.g., the level % in CV) could be posted and described in the methods.
Author Response
We appreciate the reviewer's thoughtful observations, diligent and in-depth reading of this article, and helpful suggestions that helped to elevate the work's quality.
- Most parts were improved. The assay precision and accuracy (e.g., the level % in CV) could be posted and described in the methods
Response:
Thank you for your advice . We already posted this assay in the methods.
The detection limit for this assay was 0.3 ng/mL, then repeatability values were acceptable as the intra- and inter-assay coefficients of variability (CV) were ≤ 6.4 and 6.95 %, respectively and the assay was linear on dilution. A total of 54 serum samples were tested for the comparison of the utilized kits against the commercially available validated kits and yielded acceptable correlation value (r = 0.829) using the linear regression curve.
Reviewer 2 Report
The authors attended to all the comments, however, more care should be taken in the syntax. I think the manuscript can be further improved by editing figures (extra figures on page 4) and text.
Author Response
We appreciate your helpful feedback on our manuscript. We carefully considered your feedback before making a significant correction, and we sincerely hope you will approve of it.
- The authors attended to all the comments; however, more care should be taken in the syntax. I think the manuscript can be further improved by editing figures (extra figures on page 4) and text.
Response:
- Our manuscript has been English edited by biomedicine editing services and we believe that now is in a good shape. (Already we have now a certificate which will be attached to the file).
- We apologize about this point and we removed extra figures on page 4.